# The PCNA unloader Elg1 promotes recombination at collapsed replication forks in fission yeast

**Sanjeeta Tamang, Anastasiya Kishkevich, Carl A Morrow[†], Fekret Osman, Manisha Jalan[‡], Matthew C Whitby***

Department of Biochemistry, University of Oxford, Oxford, United Kingdom

**Abstract** Protein-DNA complexes can impede DNA replication and cause replication fork collapse. Whilst it is known that homologous recombination is deployed in such instances to restart replication, it is unclear how a stalled fork transitions into a collapsed fork at which recombination proteins can load. Previously we established assays in *Schizosaccharomyces pombe* for studying recombination induced by replication fork collapse at the site-specific protein-DNA barrier *RTS1* (Nguyen et al., 2015). Here, we provide evidence that efficient recruitment/retention of two key recombination proteins (Rad51 and Rad52) to *RTS1* depends on unloading of the polymerase sliding clamp PCNA from DNA by Elg1. We also show that, in the absence of Elg1, reduced recombination is partially suppressed by deleting *fbh1* or, to a lesser extent, *srs2*, which encode known anti-recombinogenic DNA helicases. These findings suggest that PCNA unloading by Elg1 is necessary to limit Fbh1 and Srs2 activity, and thereby enable recombination to proceed.
DOI: https://doi.org/10.7554/eLife.47277.001

*For correspondence:
matthew.whitby@bioch.ox.ac.uk

Present address: [†]Department of Oncology, Weatherall Institute of Molecular Medicine, University of Oxford, Oxford, United Kingdom; [‡]Department of Radiation Oncology, Memorial Sloan Kettering Cancer Center, New York, United States

Competing interests: The authors declare that no competing interests exist.

## Introduction

Replication failure is a major cause of cancer and other pathological states through loss of genome integrity and catastrophic chromosomal rearrangements. Major hurdles to the successful completion of DNA replication include protein-DNA complexes that stall replication forks by physically impeding the Cdc45-MCM2-7-GINS (CMG) replicative helicase (*Lambert and Carr, 2013*). In some cases CMG overcomes stalling, with the help of an additional helicase and the Fork Protection Complex, with the latter ensuring that the replication proteins at the fork (collectively known as the replisome) are maintained in a state that is competent to resume DNA synthesis (*Azvolinsky et al., 2006*; *Errico and Costanzo, 2012*; *Ivessa et al., 2003*; *Leman and Noguchi, 2012*; *Sabouri et al., 2012*; *Sofueva et al., 2011*; *Steinacher et al., 2012*). However, despite these measures, replication forks can be inhibited from catalysing further DNA synthesis through a process termed fork collapse (*Cortez, 2015*). This poorly understood process involves remodelling or disassembly of the replisome and sometimes DNA breakage.

At some strong replication fork barriers (RFBs), such as *RTS1* in fission yeast, fork collapse appears to be an inevitable consequence of replication fork stalling, and results in the recruitment of homologous recombination (HR) proteins that restart replication (*Lambert et al., 2010*; *Nguyen et al., 2015*). This recombination-dependent replication (RDR) is thought to be important for ensuring the timely completion of genome duplication, helping to avoid chromosome breakage and missegregation that could otherwise occur during mitosis.

A central step in RDR is the invasion of a duplex DNA by a homologous single-stranded DNA (ssDNA) catalysed by the HR protein Rad51 (*Anand et al., 2013*). This reaction forms a displacement (D)-loop at which replication proteins are thought to reassemble (*Lydeard et al., 2010*). Rad52 aids this process by mediating the loading of Rad51 onto ssDNA coated with the ssDNA binding protein

RPA (*Krogh and Symington, 2004*). It also helps protect Rad51-ssDNA filaments from disruption by the anti-recombinogenic DNA helicases Srs2 and Fbh1 (*Lorenz et al., 2009*; *Ma et al., 2018*; *Osman et al., 2005*).

In our earlier work we showed that Rad51 and Rad52 are recruited to *RTS1* within minutes of replication fork stalling, giving rise to restarted replication that is prone to template switching (*Jalan et al., 2019*; *Nguyen et al., 2015*). However, little is known about what steps are required for the stalled replication fork to transition into a collapsed fork at which Rad51 and Rad52 can efficiently load. Presumably some disassembly and/or re-organization of the replisome is required so that HR proteins can gain access to the DNA. One of the core components of the replisome is the homotrimeric ring-shaped complex PCNA, which acts as a sliding clamp for the DNA polymerases, and scaffold for the dynamic recruitment of various proteins that promote replication and repair (*Choe and Moldovan, 2017*). As PCNA encircles DNA it has to be actively unloaded from chromosomes following both the completion of each Okazaki fragment and termination of replication. However, it is unknown whether PCNA has to be unloaded for recombination to occur at a stalled/collapsed replication fork.

A principle factor for unloading PCNA is Elg1 (ATAD5 in humans) (*Kubota et al., 2013a*; *Lee et al., 2013*). Elg1/ATAD5 forms a replication factor C (RFC)-like complex with Rfc2-5 (*Bellaoui et al., 2003*; *Ben-Aroya et al., 2003*; *Kanellis et al., 2003*), which is vital for genome stability, and, in mice and humans, appears to act as a tumour suppressor (*Bell et al., 2011*; *Gazy et al., 2015*; *Johnson et al., 2016*; *Maleva Kostovska et al., 2016*; *Shemesh et al., 2017*; *Sikdar et al., 2009*). Here we discover that fission yeast lacking Elg1 exhibit reduced levels of *RTS1*-induced recombination and fewer cells with Rad51 and Rad52 foci colocalizing with *RTS1*. We also show that the hypo-recombination in an *elg1Δ* mutant is fully suppressed by a mutation rendering PCNA prone to disassembly, and partially suppressed by deleting either *fbh1* or *srs2*. These findings suggest that PCNA unloading by Elg1, during replication fork collapse, is necessary to temper Fbh1 and Srs2 activity so that recombination can proceed.

## Results and discussion

### Elg1 promotes *RTS1*-induced recombination

We have previously described a system for studying replication fork collapse and restart using the site-specific protein-DNA RFB *RTS1* in the fission yeast *Schizosaccharomyces pombe* (*Ahn et al., 2005*; *Jalan et al., 2019*; *Nguyen et al., 2015*). In our standard assay, *RTS1* is inserted between a direct repeat of mutant *ade6⁻* heteroalleles on chromosome 3 (the '0 kb site') so that recombination can be measured by determining the frequency of two types of *ade6⁺* recombinants (gene conversions and deletions) (*Figure 1A,B*). As *RTS1* is a polar RFB, and the *ade6* locus is replicated with a strong directional bias (telomere to centromere), only one orientation of the barrier blocks forks at this genomic location, which we refer to as the active orientation (AO) (*Nguyen et al., 2015*). The opposite orientation, which does not block replication, is called the inactive orientation (IO). A comparison of the frequency of *ade6⁺* recombinants in strains with and without *RTS1-IO* shows that the inactive barrier has no effect on the frequency of recombination (*Jalan et al., 2019*; *Nguyen et al., 2015*). In contrast, *RTS1-AO* strongly induces recombination (*Nguyen et al., 2015*) (*Figure 1C,D*).

To investigate whether *RTS1*-induced recombination depends on the PCNA unloader Elg1, we compared recombination frequencies in wild-type and *elg1Δ* strains harbouring either *RTS1-IO* or -AO (*Figure 1C,D*). In line with the observation that an *elg1Δ* mutant exhibits increased spontaneous direct repeat recombination in budding yeast (*Gazy et al., 2013*), a fission yeast *elg1Δ* mutant containing *RTS1-IO* exhibits slightly higher levels of spontaneous recombination than a wild-type strain with *RTS1-IO*, suggesting that Elg1 has an anti-recombinogenic function (*Figure 1C*). However, this modest increase in recombination is not observed in all genetic backgrounds (see Figure 3A). In stark contrast to the modest increase in spontaneous recombination, the elevated levels of recombination in wild-type strains containing *RTS1-AO* are reduced dramatically in an *elg1Δ* mutant, with gene conversions down ~20 fold and deletions down ~4 fold (*Figure 1D*). These reductions are not caused by any weakening of the *RTS1* barrier, as native two-dimensional gel electrophoretic (2DGE)

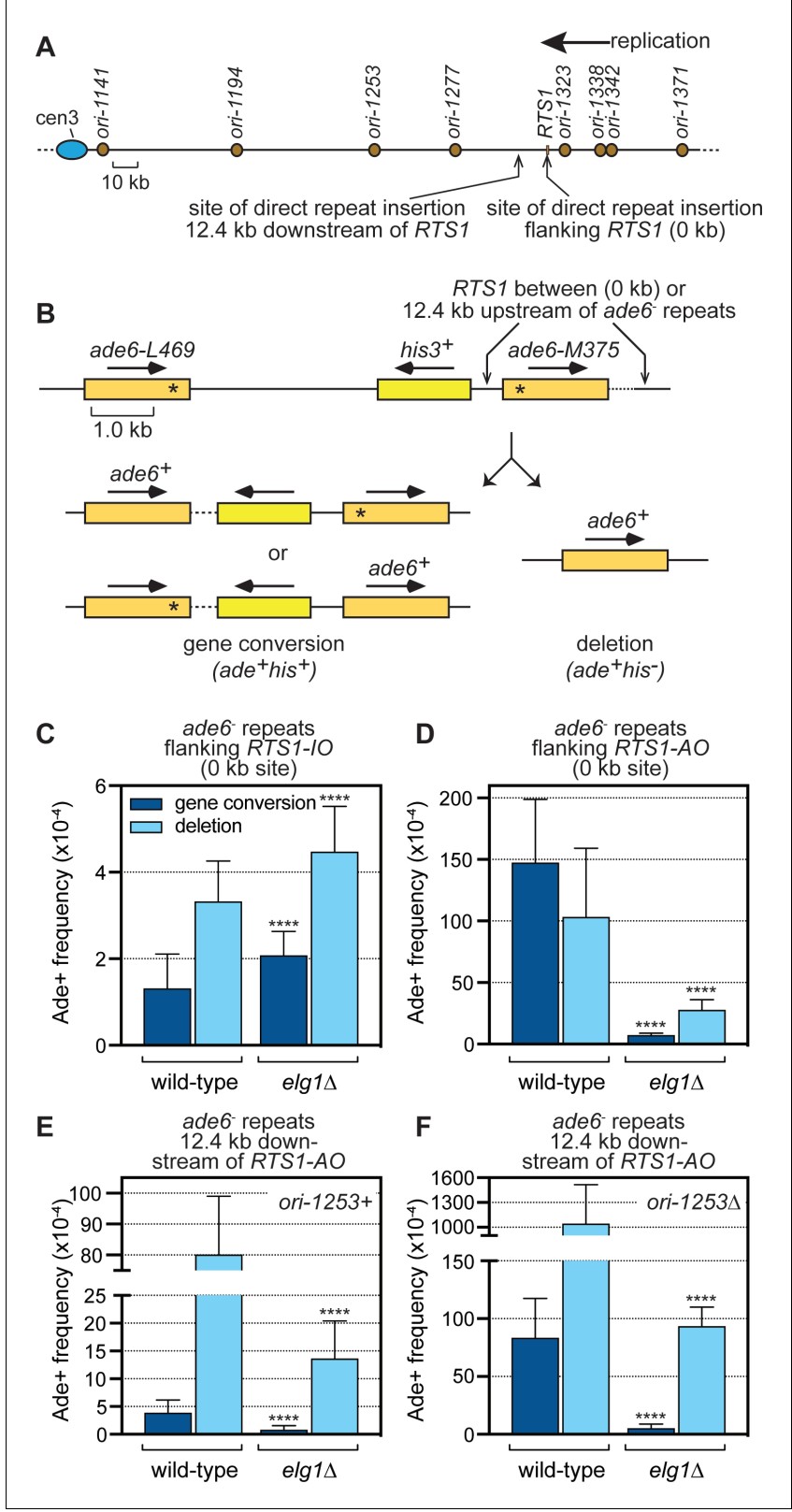

**Figure 1.** Spontaneous (*RTS1-IO*) and *RTS1-AO*-induced direct repeat recombination in wild-type and *elg1Δ* strains. (**A**) Map showing the position of *RTS1*, origins of replication and sites of insertion for the direct repeat recombination reporter on chromosome 3. (**B**) Schematic of the direct repeat recombination reporter showing the '0 kb' and '12.4 kb' *RTS1* sites and two types of Ade⁺ recombinant. Asterisks indicate the position of point

*Figure 1 continued on next page*

*Figure 1 continued*

mutations in *ade6-L469* and *ade6-M375*. (**C**) Ade$^+$ recombinant frequencies for strains MCW4712 and MCW7706. (**D**) Ade$^+$ recombinant frequencies for strains MCW4713 and MCW7708. (**E**) Ade$^+$ recombinant frequencies for strains MCW7259 and MCW8191. (**F**) Ade$^+$ recombinant frequencies for strains MCW7295 and MCW8290. Data are mean values with error bars showing 1 SD. Significant differences relative to equivalent wild-type values are indicated by *p<0.05, **p<0.01, ***p<0.001, ****p<0.0001. Ade$^+$ recombinant frequencies with statistical analysis are also shown in **Supplementary file 1**.

DOI: https://doi.org/10.7554/eLife.47277.002

The following figure supplement is available for figure 1:

**Figure supplement 1.** Elg1 is not required for replication fork blocking at *RTS1-AO*.

DOI: https://doi.org/10.7554/eLife.47277.003

analysis of replication intermediates shows similar levels of blocked replication forks in wild-type and *elg1Δ* strains (**Figure 1—figure supplement 1**). Previous studies in budding yeast have shown that loss of Elg1 generally causes increased spontaneous recombination (**Ben-Aroya et al., 2003**), however, recombination induced by methylmethane sulfonate (MMS) and phleomycin, which cause lesions that can impede replication forks, was reported to be reduced in an *elg1Δ* mutant (**Ogiwara et al., 2007**). Our data show that Elg1, in addition to having an anti-recombinogenic function, also has a pro-recombinogenic function that is specifically required to promote RFB-induced recombination, which likely explains the reduction in MMS/phleomycin-induced recombination observed previously in budding yeast.

## Template switch recombination downstream of *RTS1-AO* is strongly reduced in an *elg1Δ* mutant

RDR, triggered by *RTS1-AO*, exhibits a high frequency of template switching (TS) downstream of the barrier, which can be measured using the *ade6*$^-$ direct repeat recombination reporter (**Jalan et al., 2019**; **Nguyen et al., 2015**) (**Figure 1A,B**). To see whether Elg1 is required for TS, we compared the frequency of Ade$^+$ recombinants in wild-type and *elg1Δ* strains with the reporter positioned 12.4 kb downstream of *RTS1-AO* (the '12.4 kb site') (**Figure 1E**). Gene conversions and deletions were reduced in an *elg1Δ* mutant by ~5 fold and ~6 fold, respectively, with the former being suppressed to spontaneous levels. These data show that Elg1 promotes TS associated with restarted replication.

There are several non-mutually exclusive ways in which loss of Elg1 could lead to a reduction in TS: 1) enhanced stability of restarted replication resulting in less frequent TS events; 2) slower TS kinetics resulting in fewer completed events prior to convergence with an oncoming replication fork; and 3) less efficient RDR resulting in fewer cells with restarted replication reaching the downstream reporter prior to convergence with an oncoming replication fork. We have previously shown that, even in wild-type cells, only a minority of restarted replication reaches the reporter 12.4 kb downstream of *RTS1-AO* to register a TS event (**Nguyen et al., 2015**). We have also shown that deleting the strongest centromere-proximal origin (*ori-1253*) provides more time for RDR to reach the downstream reporter leading to a dramatic increase in TS (**Jalan et al., 2019**; **Nguyen et al., 2015**). We reasoned that if the reduction in TS in an *elg1Δ* mutant was solely due to an increase in restarted replication stability, then the fold increase in recombination upon *ori-1253* deletion should be the same in both wild-type and *elg1Δ* mutant. In line with previous data (**Nguyen et al., 2015**), deletion of *ori-1253* in the wild-type resulted in a ~21 fold increase in gene conversions and a ~13 fold increase in deletions (**Figure 1E,F**). In contrast, the increase in an *elg1Δ* mutant was only ~6 fold for gene conversions and ~7 fold for deletions (**Figure 1E,F**). Whilst not discounting the possibility that Elg1 affects restarted replication stability, these data suggest that Elg1 promotes the kinetics of TS and/or efficient RDR. However, a deficiency in RDR is not evident from our 2DGE analysis of replication intermediates at *RTS1-AO*, as both wild-type and *elg1Δ* mutant exhibit similar levels of large Y-shaped and double Y-shaped DNA molecules (representative of replication past the barrier, and fork convergence at the barrier, respectively) (**Figure 1—figure supplement 1B,C**). By a process of elimination, these data suggest that Elg1 is required for the TS process itself, although this conclusion assumes that 2DGE analysis provides an accurate measure of RDR efficiency, which may not always be true (see below).

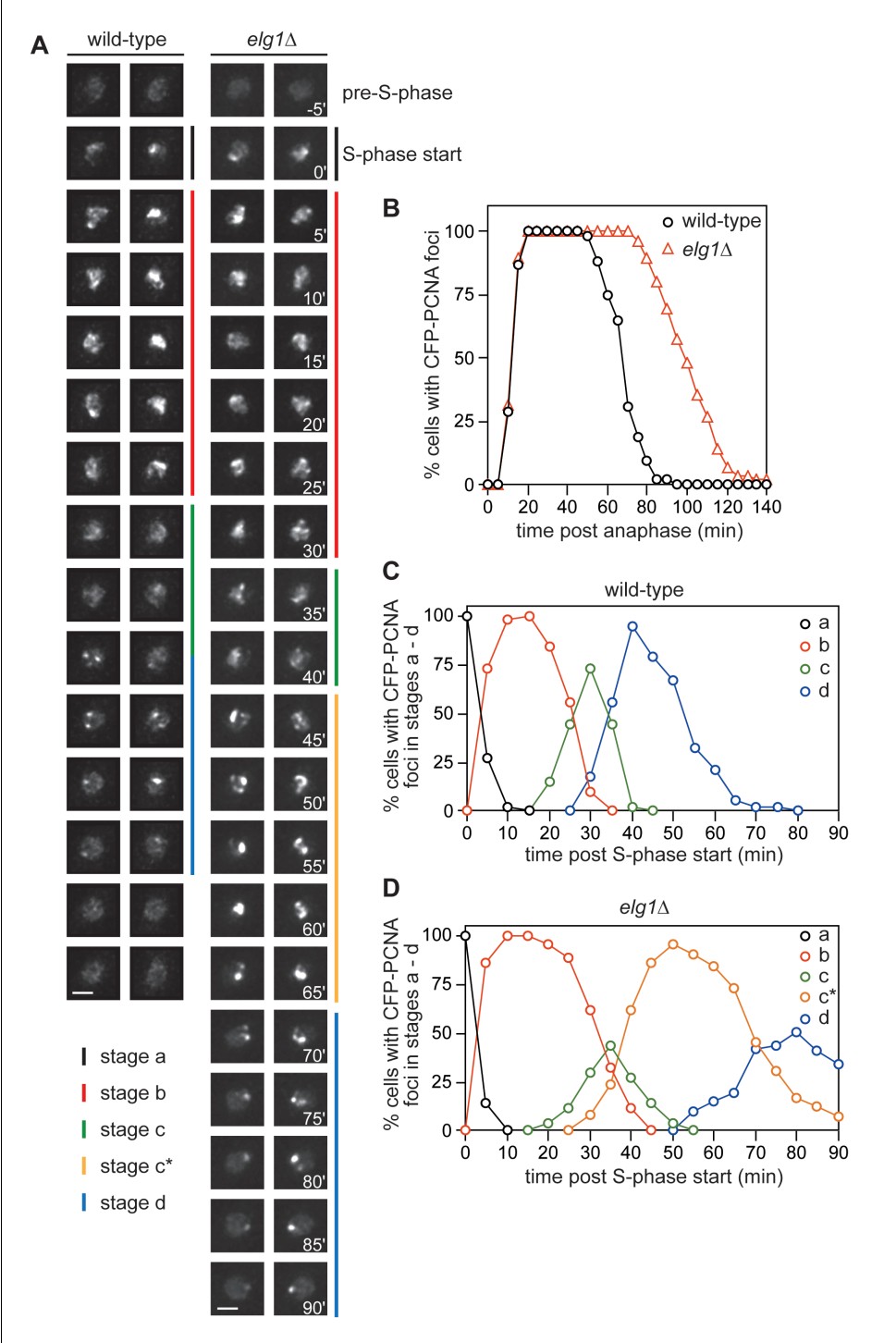

**Figure 2.** Comparison of CFP-PCNA fluorescence in wild-type and *elg1Δ* strains. (A) Stills taken from four separate time-lapse movies (two each for wild-type and *elg1Δ*) showing the different stages of nuclear CFP-PCNA fluorescence at 5 min intervals during representative cell cycles. Scale bar = 2 μm. (B) The percentage of cells with nuclear CFP-PCNA fluorescence above the pre-S-phase level at the indicated times post anaphase. (C – D) The percentage of cells with nuclear CFP-PCNA fluorescence in stages a – d. The data in B – D are taken from time-lapse movies like those shown in A. The wild-type strain is MCW7065 (n = 52) and the *elg1Δ* strain is MCW7965 (n = 85).

DOI: https://doi.org/10.7554/eLife.47277.004

2DGE analysis does not provide a definitive measure of replication restart efficiency because the accumulation of replication intermediates, within the DNA restriction fragment being analysed, depends on both their frequency and half-life within the cell population. For example, a reduction in velocity of restarted replication could offset the effect of a reduced frequency of restarted forks (i.e. even though fewer cells within a population contain a restarted fork, those that do are more likely to contain one that resides within the restriction fragment at the point of analysis, resulting in little or no apparent change in the amount of large Y-shaped DNA molecules that are detected). Similarly, an increase in the frequency of fork convergence at the RFB (which would normally lead to a relative increase in large Y-shaped DNA molecules) could be offset by an increase in the speed with which converging forks are processed. Therefore, the absence of any apparent difference between wild-type and *elg1Δ* mutant in the accumulation of replication intermediates at around *RTS1-AO* (*Figure 1—figure supplement 1B,C*), does not rule out the possibility that RDR is deficient in an *elg1Δ* mutant (e.g. if fork convergence is faster, and restarted replication is slower, in an *elg1Δ* mutant, then this could result in no apparent change in the relative amount of large Y-shaped and double Y-shaped DNA molecules compared to wild-type). Taking this into account, we cannot exclude the possibility that Elg1 promotes RDR and, in so-doing, drives TS at the 12.4 kb reporter.

## Removal of PCNA from DNA at the end of S-phase is delayed in the absence of Elg1

Before investigating whether Elg1 promotes *RTS1*-induced recombination by catalysing PCNA unloading, we first wanted to confirm that Elg1 is required for PCNA unloading in fission yeast. The spatiotemporal dynamics of DNA replication in living *S. pombe* cells can be followed using a strain in which PCNA is tagged with enhanced cyan fluorescent protein (CFP) (*Meister et al., 2007*). This strain also contains untagged PCNA, which compensates for slight deficiencies in the tagged form of the protein. Four characteristic patterns of CFP-PCNA nuclear fluorescence during the cell cycle have been described (*Meister et al., 2007*). To see if these patterns are altered in an *elg1Δ* mutant we used time-lapse microscopy, in which images of cells were taken every 5 min, and then staged relative to anaphase. In wild-type cells we observed similar patterns of CFP-PCNA fluorescence as documented previously (*Meister et al., 2007*). Prior to S-phase the fluorescence is weak and diffuse, and occupies mainly the non-nucleolar part of the nucleus (*Figure 2A*). Between 10–15 min post anaphase bright foci appear marking the start of S-phase (pattern a), which within ~5 min spread to fill much of the extranucleolar space (pattern b) (*Figure 2A,B,C*). After ~20–25 min the bright fluorescence resolves into a slightly dimmer and more punctate pattern with fewer foci (pattern c), and then after another ~15–20 min transitions to only one or two foci at the edges of the nucleolus (pattern d) (*Figure 2A,B,C*). By ~60 min CFP-PCNA fluorescence returns to its pre-S-phase level. In an *elg1Δ* mutant patterns a – c appear similar to wild-type, albeit patterns b and c persist slightly longer (*Figure 2A,B,D*). However, unlike wild-type, pattern c does not transition into pattern d, instead ~30–40 min after the start of S-phase a new pattern of fluorescence emerges, which is characterised by large patches of very bright fluorescence (pattern c*). These persist for up to ~40 min before changing into something that is more akin to pattern d (*Figure 2A,D*). Altogether these data show that the start of S-phase is not delayed in an *elg1Δ* mutant. However, the persistence of PCNA foci at later time points in *elg1Δ* mutant cells suggests a deficiency in PCNA removal from DNA, and is consistent with previous observations in both yeast and mammalian cells (*Etheridge et al., 2014*; *Kubota et al., 2013b*; *Shiomi and Nishitani, 2013*).

## A disassembly-prone mutant of PCNA suppresses *elg1Δ* hypo-recombination

If Elg1's role in promoting *RTS1*-induced recombination is to catalyse PCNA unloading, then mutations at the trimer interface of PCNA, which allow it to more readily dissociate from DNA, might suppress *elg1Δ* hypo-recombination. One such mutation is D150E, which rescues MMS sensitivity, hyper-sister chromatid recombination and telomere elongation in a budding yeast *elg1Δ* mutant (*Johnson et al., 2016*). We introduced the D150E mutation into the fission yeast PCNA gene (*pcn1*) together with a genetically linked selectable marker (*natMX4*). The mutant strain was viable and exhibited similar growth to the wild-type parental strain, and to a *pcn1*+ strain with the same *natMX4* linked marker (data not shown). We next compared the frequency of recombination at the 0 kb site

in the $pcn1^+$ and $pcn1^{D150E}$ strains with RTS1-IO and -AO in $elg1^+$ and $elg1\Delta$ backgrounds (*Figure 3*). Spontaneous recombination was increased 2–3-fold in the $pcn1^+$ strain compared to our standard wild-type strain (compare data in *Figures 1C* and *3A*), suggesting that the presence of the linked *natMX4* marker might be affecting *pcn1* expression. Interestingly, the $pcn1^{D150E}$ mutant exhibited an even higher frequency of recombination, which was 2–3-fold more than the $pcn1^+$ strain (*Figure 1C*). This suggests that proper retention of PCNA on DNA is important for suppressing spontaneous recombination. In both $pcn1^+$ and $pcn1^{D150E}$ strains, deletion of *elg1* had no effect on the frequency of spontaneous recombination (*Figure 3A*). However, similar to the data in *Figure 1D*, deletion of *elg1* in a $pcn1^+$ strain containing RTS1-AO results in a ~9 fold decrease in

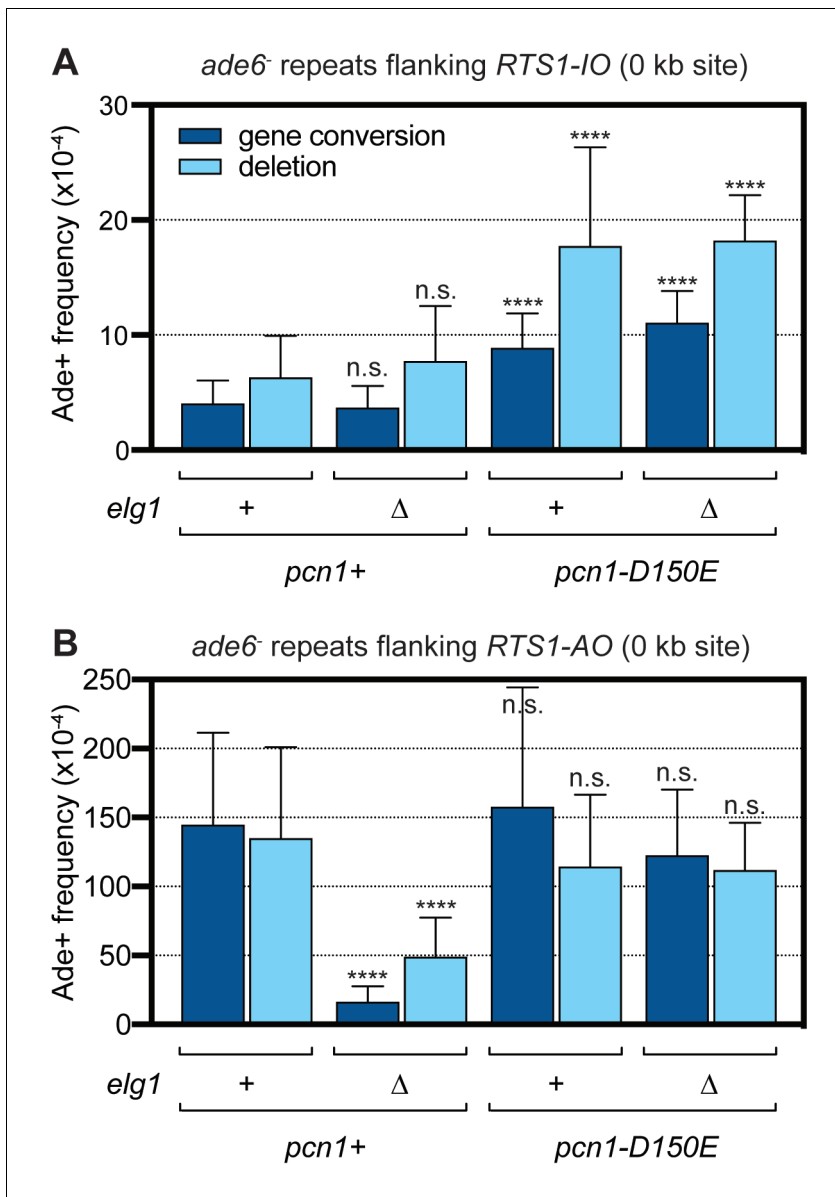

**Figure 3.** Spontaneous (*RTS1-IO*) and *RTS1-AO*-induced direct repeat recombination in $pcn1^+$ and $pcn1^{D150E}$ strains with and without *elg1*. (**A**) Ade⁺ recombinant frequencies for strains MCW9394, MCW9390, MCW9183 and MCW9187. (**B**) Ade⁺ recombinant frequencies for strains MCW9396, MCW9392, MCW9185 and MCW9189. Data are mean values with error bars showing 1 SD. Significant differences relative to equivalent $elg1^+$ $pcn1^+$ values are indicated by *p<0.05, **p<0.01, ***p<0.001, ****p<0.0001. n.s. = not significant. Ade⁺ recombinant frequencies with statistical analysis are also shown in *Supplementary file 1*.
DOI: https://doi.org/10.7554/eLife.47277.005

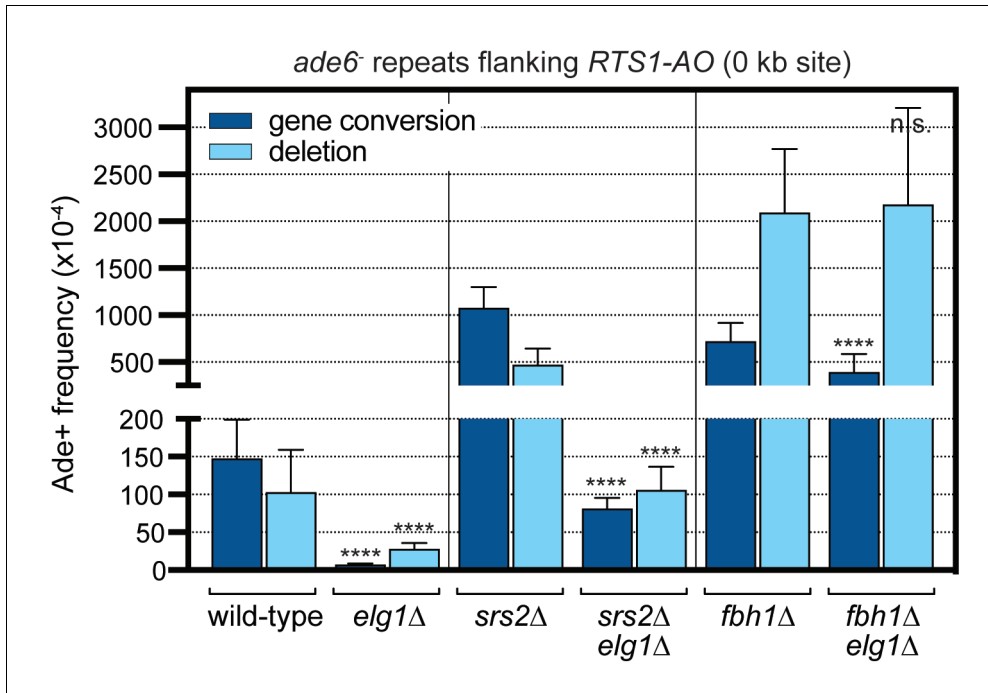

**Figure 4.** Effect of deleting *srs2* and *fbh1* on the frequency of *RTS1-AO*-induced direct repeat recombination in wild-type and *elg1Δ* strains. The strains are MCW4713, MCW7708, FO1750, MCW8330, FO1816 and MCW8946. Data are mean values with error bars showing 1 SD. Significant differences relative to equivalent *elg1+* strain values are indicated by \*p<0.05, \*\*p<0.01, \*\*\*p<0.001, \*\*\*\*p<0.0001. n.s. = not significant. Ade[+] recombinant frequencies with statistical analysis are also shown in ***Supplementary file 1***.
DOI: https://doi.org/10.7554/eLife.47277.006

The following figure supplement is available for figure 4:

**Figure supplement 1.** Growth and genotoxin sensitivities of *elg1Δ*, *srs2Δ* and *fbh1Δ* single and double mutants.
DOI: https://doi.org/10.7554/eLife.47277.007

gene conversions and a ~3 fold decrease in deletions (***Figure 3B***). In contrast, deleting *elg1* in a *pcn1[D150E]* strain causes no reduction in recombination (***Figure 3B***). These data strongly suggest that Elg1 promotes *RTS1*-induced recombination through its ability to unload PCNA from DNA.

## Srs2 is partly responsible for the reduction in *RTS1*-induced recombination in an *elg1Δ* mutant

In budding yeast, the Srs2 DNA helicase is recruited to stalled replication forks by SUMOylated PCNA, where it can limit recombination by disrupting Rad51-DNA filaments (***Motegi et al., 2006***; ***Papouli et al., 2005***; ***Pfander et al., 2005***). To achieve the right level of Srs2 activity, so that beneficial recombination proceeds whilst toxic recombination is kept in check, the SUMO-like domain-containing protein Esc2 directs Srs2 turnover by the SUMO targeted ubiquitin ligase Slx5-Slx8 (***Urulangodi et al., 2015***). It also helps retain Elg1 at sites of replication stress and, thereby, may limit the concentration of Srs2 at the blocked fork by promoting PCNA unloading (***Urulangodi et al., 2015***).

In fission yeast it remains unknown how Srs2 is recruited to stalled replication forks. Nevertheless, we wondered whether the unloading of PCNA by Elg1 might limit Srs2 activity at the blocked fork. To investigate this possibility, we compared *RTS1*-induced recombination in a *srs2Δ* single mutant and *srs2Δ elg1Δ* double mutant, reasoning that if Elg1's role in promoting recombination is to limit Srs2 activity, then removal of Srs2 should obviate its need (i.e. a *srs2Δ elg1Δ* double mutant should exhibit the same frequency of recombination as a *srs2Δ* single mutant). However, if Elg1's pro-recombinogenic role is independent of Srs2, then the fold reduction in recombination frequency between *srs2Δ* single mutant and *srs2Δ elg1Δ* double mutant should be the same as that between

wild-type and *elg1Δ* single mutant. Consistent with our previous data, a *srs2Δ* mutant exhibits greatly increased levels of *RTS1*-induced recombination (*Figure 4*) (*Jalan et al., 2019*; *Lorenz et al., 2009*). This is reduced in a *srs2Δ elg1Δ* double mutant with gene conversions decreasing by ~13 fold and deletions by ~4 fold (*Figure 4*). This fold reduction in deletions is approximately the same as in a *srs2+* background, however, the reduction in gene conversions is noticeably less (~13 fold versus ~20 fold). This suggests that Srs2 is partly responsible for the reduction in gene conversions in an *elg1Δ* mutant. However, the fact that much of the hyper-recombination in a *srs2Δ* mutant is still suppressed by deletion of *elg1* indicates that Elg1's pro-recombinogenic function is mostly independent of limiting Srs2 activity.

In budding yeast, a *srs2Δ elg1Δ* double mutant exhibits impaired growth and a synergistic increase in sensitivity to DNA damaging agents, including MMS and the Topoisomerase I inhibitor camptothecin (CPT), which causes replication fork breakage (*Gazy et al., 2013*; *Parnas et al., 2010*). In contrast, a fission yeast *srs2Δ elg1Δ* double mutant, although exhibiting a synergistic increase in sensitivity to MMS, exhibits no obvious deficiency in growth and a CPT sensitivity that is the same as a *srs2Δ* single mutant (*Figure 4—figure supplement 1A*). Therefore, the differences in recombination between single and double mutant strains is unlikely to stem from a difference in their growth and viability.

## Fbh1 is responsible for most of the reduction in *RTS1*-induced recombination in an *elg1Δ* mutant

In addition to Srs2, fission yeast harbours another related anti-recombinogenic DNA helicase called Fbh1, which is absent in budding yeast but conserved in humans (*Chiolo et al., 2007*; *Morishita et al., 2005*; *Osman et al., 2005*). Similar to Srs2, Fbh1 limits Rad51-dependent recombination at blocked replication forks by disrupting Rad51-DNA filaments (*Lorenz et al., 2009*; *Simandlova et al., 2013*; *Tsutsui et al., 2014*). It is also part of an E3 ligase complex that ubiquitinates Rad51 to target it for degradation (*Chu et al., 2015*; *Tsutsui et al., 2014*). In humans, Fbh1 is recruited to sites of DNA replication and damage by interaction with PCNA mediated by two PCNA-binding motifs (an N-terminal PIP-box and C-terminal APIM) (*Bacquin et al., 2013*). Even though these PCNA-binding motifs are not conserved in fission yeast Fbh1, we still decided to investigate whether Fbh1 is responsible for the reduction in *RTS1*-induced recombination in an *elg1Δ* mutant. First, we assessed the viability and genotoxin sensitivity of an *fbh1Δ elg1Δ* double mutant (*Figure 4—figure supplement 1B*). An *fbh1Δ* single mutant exhibits impaired growth relative to wild-type (*Osman et al., 2005*), which is not exacerbated by deletion of *elg1*. However, an *fbh1Δ elg1Δ* double mutant does exhibit a synergistic increase in sensitivity to both MMS and CPT.

Having established that an *fbh1Δ elg1Δ* double mutant is viable, we applied the same logic used when testing Srs2's involvement in the *elg1Δ* hypo-recombination phenotype, and compared the frequency of *RTS1*-induced recombination in an *fbh1Δ* single mutant and *fbh1Δ elg1Δ* double mutant (*Figure 4*). Remarkably, the hyper-recombination of an *fbh1Δ* mutant was only slightly reduced by deletion of *elg1Δ*, with the frequency of deletions remaining the same in both single and double mutant, and gene conversions decreasing by only ~1.8 fold. These data suggest that Fbh1 is responsible for most of the reduction in *RTS1*-induced recombination observed in an *elg1Δ* mutant.

We have shown previously that overexpressing Fbh1, with its helicase activity intact, is sufficient to suppress *RTS1*-induced recombination and, interestingly, the fold reduction in deletions is the same as in an *elg1Δ* mutant (~4 fold), with the reduction in gene conversions being slightly less (~7 fold versus ~20 fold in an *elg1Δ* mutant) (*Lorenz et al., 2009*). These observations suggest that Elg1 could promote *RTS1*-induced recombination by restricting the localised concentration of Fbh1 at the barrier. Presumably Fbh1 activity at the barrier is somehow dependent on PCNA, however, so far, we have been unable to detect an interaction between these proteins by co-immunoprecipitation (data not shown). Interestingly, the Pif1 family helicase Pfh1, which promotes fork convergence and RDR at *RTS1* (*Jalan et al., 2019*; *Steinacher et al., 2012*), interacts with both PCNA and Fbh1 (*McDonald et al., 2016*). It is therefore possible that Fbh1's recruitment and/or retention to sites of replication fork collapse is mediated by Pfh1, which, in turn, depends on PCNA.

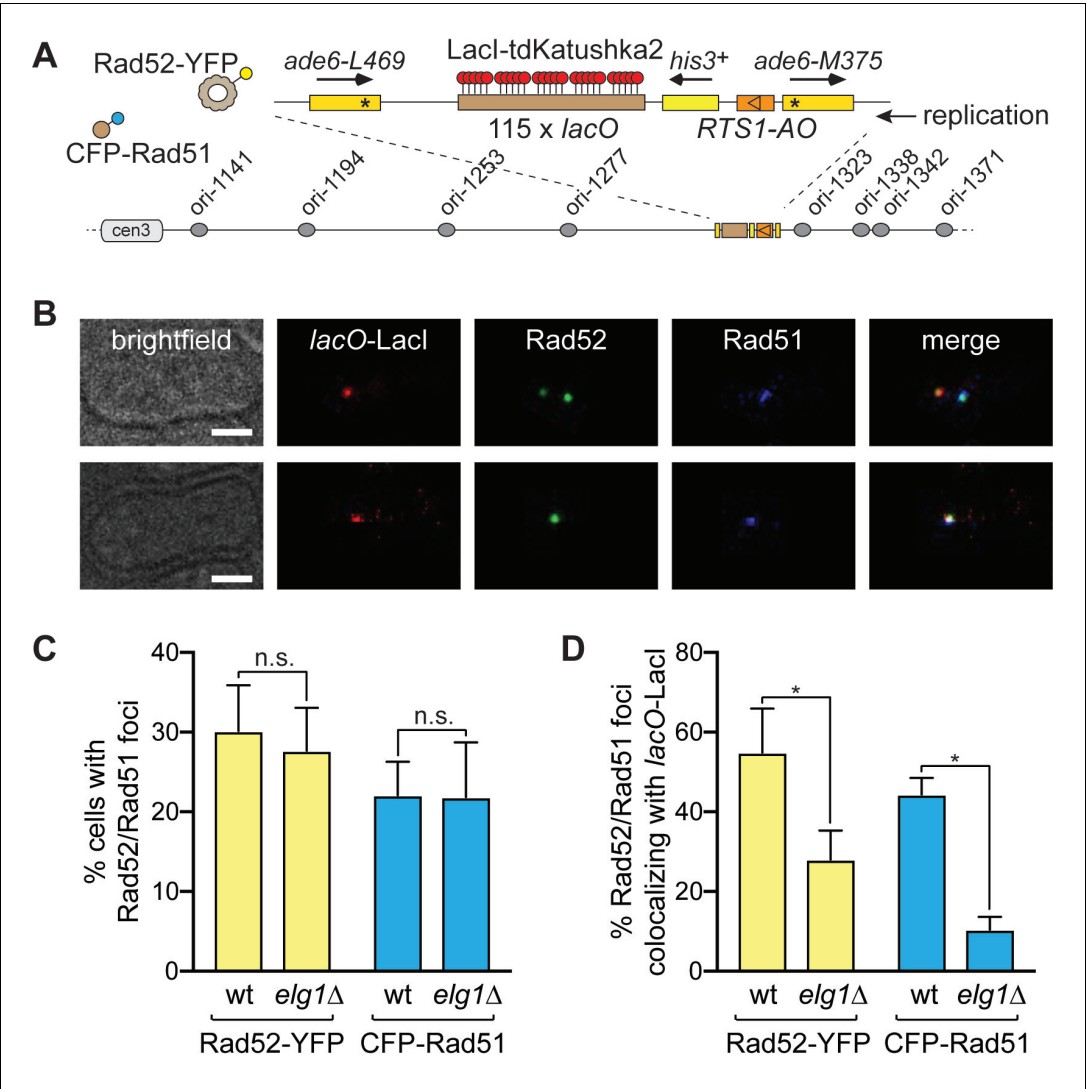

**Figure 5.** CFP-Rad51 and Rad52-YFP foci colocalization with *RTS1-AO* in wild-type and *elg1Δ* strains.  (A) Schematic showing key components of the strains used for imaging, including the modification of the 0 kb site recombination reporter with a 115 repeat *lacO* array. (B) Examples of two wild-type cells with CFP-Rad51 and Rad52-YFP foci. In the top panels, the cell contains two Rad52-YFP foci, one of which co-localizes with a *lacO*-LacI-tdKatushka2 focus, which marks the *RTS1-AO* location, and the second co-localizes with a CFP-Rad51 focus. The bottom panels show a cell with CFP-Rad51 and Rad52-YFP foci co-localizing with a *lacO*-LacI-tdKatushka2 focus. The scale bar = 2 μm. (C) Percentage of wild-type and *elg1Δ* cells with Rad52-YFP and CFP-Rad51 foci. Note that the values for wild-type are higher than observed previously (*Nguyen et al., 2015*) due to a greater proportion of S-phase and early G2 cells in the cultures that were analysed (data not shown). (D) Percentage of Rad52-YFP/CFP-Rad51 foci positive cells containing a Rad52-YFP and/or CFP-Rad51 focus that co-localizes with *lacO*-LacI-tdKatushka2. Data in (C) and (D) are mean values from four independent experiments (~100 cells were analysed in each experiment). Error bars show 1 SD. Significant differences between wild-type and *elg1Δ* are indicated by *p<0.05. n.s. = not significant. The strains are MCW7638 (wild-type) and MCW8921 (*elg1Δ*).
DOI: https://doi.org/10.7554/eLife.47277.008

## Recruitment/retention of Rad51 and Rad52 to *RTS1* is reduced in an *elg1Δ* mutant

As both Fbh1 and Srs2 suppress recombination by disrupting Rad51-DNA filaments (*Krejci et al., 2003*; *Simandlova et al., 2013*; *Tsutsui et al., 2014*; *Veaute et al., 2003*), we reasoned that if they are responsible for the hypo-recombination in an *elg1Δ* mutant, then the amount of Rad51 at the

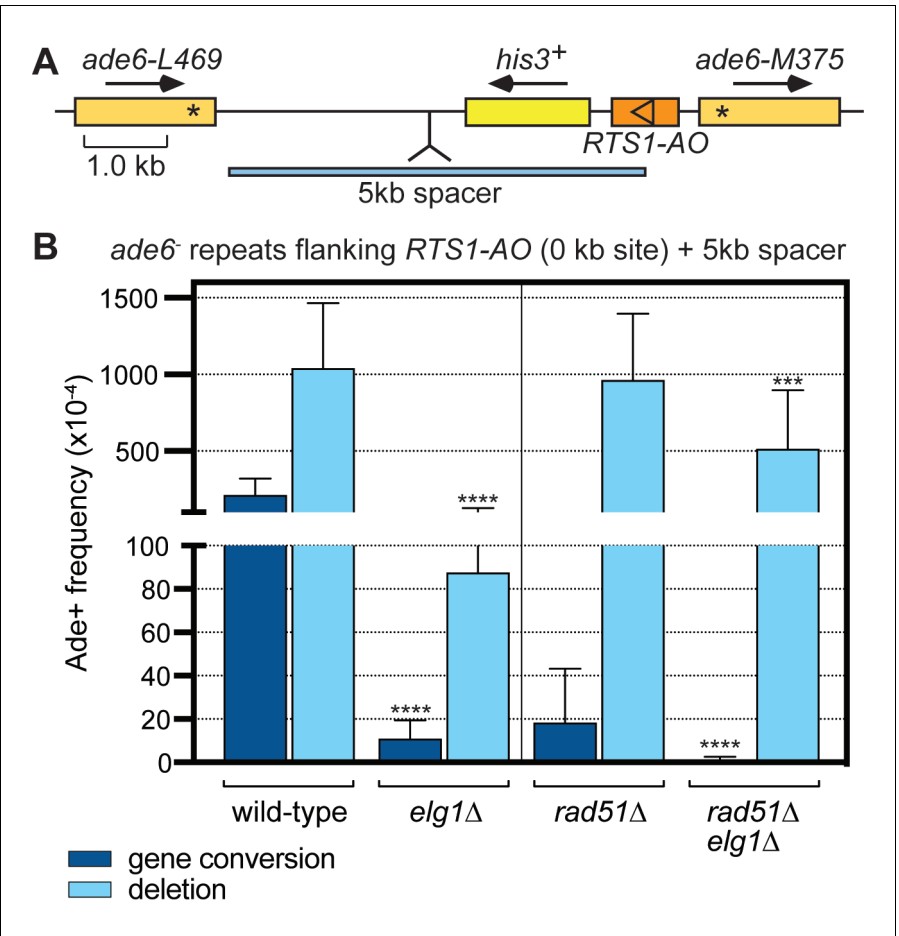

**Figure 6.** Elg1 promotes IFSA in the presence and absence of Rad51. (**A**) Schematic showing the insertion of the 5 kb DNA spacer at the 0 kb site recombination reporter. (**B**) Ade⁺ recombinant frequencies for strains MCW8023, MCW8941, MCW8136 and MCW8943. Data are mean values with error bars showing 1 SD. Significant differences relative to equivalent *elg1*⁺ strain values are indicated by *p<0.05, **p<0.01, ***p<0.001, ****p<0.0001. Ade⁺ recombinant frequencies with statistical analysis are also shown in *Supplementary file 1*.
DOI: https://doi.org/10.7554/eLife.47277.009

barrier should also be reduced. To investigate this, we used microscopy to compare the frequency of Rad51 foci co-localizing with *RTS1-AO* in wild-type and *elg1Δ* cells (*Figure 5*). Rad51 foci were detected by an N-terminal CFP tag, whilst the position of *RTS1* was marked by the binding of a LacI-tdKatushka2 fusion to an adjacent *lacO* array (*Nguyen et al., 2015*) (*Figure 5A,B*). Additionally, we detected Rad52 foci by means of a C-terminal yellow fluorescent protein (YFP) tag (*Figure 5A,B*). The overall percentage of cells with Rad51 and/or Rad52 foci was similar in both wild-type and *elg1Δ* mutant (*Figure 5C*). However, whereas ~44% of Rad51 foci and ~55% of Rad52 foci co-localize with *RTS1* in wild-type, in an *elg1Δ* mutant this drops to ~10% and ~28%, respectively (*Figure 5D*). These data show that Rad51 recruitment and/or retention to *RTS1* is reduced in an *elg1Δ* mutant, which is consistent with the notion that Fbh1 and Srs2 activity is enhanced. However, the reduction in Rad52 foci was unexpected as overexpression of Fbh1, whilst diminishing Rad51 foci, has no effect on Rad52 foci (*Lorenz et al., 2009*). It is possible that the reduction in Rad52 foci is due to increased Srs2 activity, as budding yeast Srs2 is capable of dislodging Rad52 from ssDNA in vitro (*De Tullio et al., 2017*).

## Elg1 promotes Rad51-independent recombination

If Elg1 is required for efficient recruitment and/or retention of Rad52 to *RTS1*, then it should be needed to promote Rad51-independent recombination at the barrier. We recently showed that

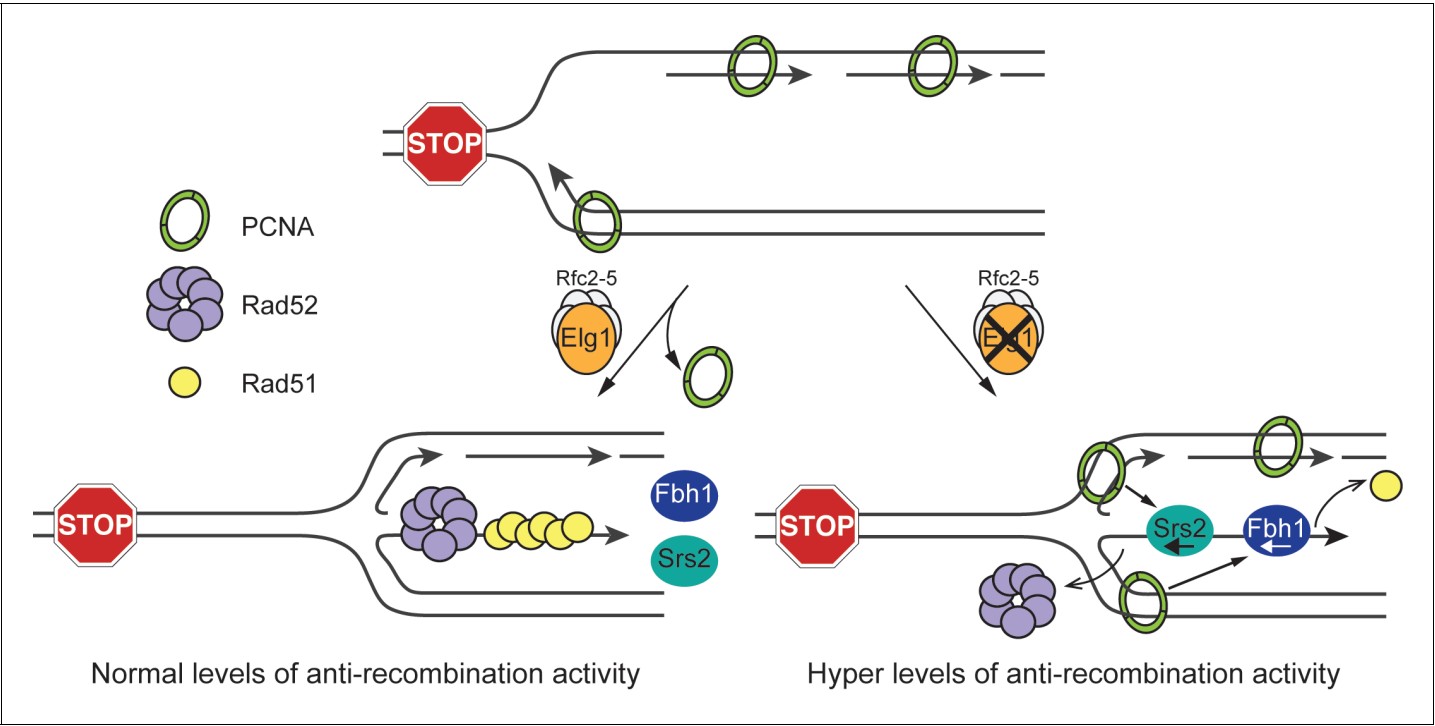

**Figure 7.** Model showing how Elg1 promotes recombination at a blocked replication fork. Upon encounter with *RTS1-AO*, the replication fork undergoes reversal forming a 'chicken foot' structure with an exposed duplex DNA end. Resection of this DNA duplex, by the nucleolytic activity of the Mre11-Rad50-Nbs1-Ctp1 complex and Exo1, generates a 3'-ended ssDNA tail onto which Rad51 loads with the help of Rad52. The loading and/or retention of Rad51 and Rad52 at the barrier is inhibited by the activities of Fbh1 and/or Srs2, which are promoted either directly or indirectly by PCNA bound to the blocked fork. Elg1 unloads PCNA from the blocked fork and thereby limits Fbh1 and Srs2 activity. In the absence of Elg1, PCNA is retained at the blocked fork leading to excessive Fbh1 and Srs2 activity and, therefore, reduced recombination.
DOI: https://doi.org/10.7554/eLife.47277.010

Rad52 can promote strand annealing, between forks converging at *RTS1*, independently of Rad51 (*Morrow et al., 2017*). This inter-fork strand annealing (IFSA) is enhanced when the spacing between the *ade6⁻* repeats flanking *RTS1* is increased. To investigate whether Elg1 is required for IFSA, in the presence and absence of Rad51, we compared the frequency of *RTS1-AO*-induced recombination in wild-type, *elg1Δ*, *rad51Δ* and *rad51Δ elg1Δ* strains containing the 0 kb site reporter with an extra 5 kb DNA spacer between the *ade6⁻* repeats (*Figure 6A,B*). Consistent with our previous data (*Morrow et al., 2017*), the frequency of deletions increases ~10 fold in wild-type compared to the equivalent strain without a 5 kb spacer, whereas gene conversions increase by only ~1.5 fold. In an *elg1Δ* mutant the frequency of gene conversions decreases by ~19 fold, which is similar to the fold reduction seen in the equivalent strain without a 5 kb spacer (compare data in *Figures 1D* and *6B*). However, the fold reduction in deletions is ~12 fold, which is noticeably more than the ~4 fold in the equivalent strain without a spacer. In the absence of Rad51, the frequency of deletions remains essentially unchanged, whereas gene conversions are reduced by ~12 fold. Importantly, in a *rad51Δ elg1Δ* double mutant, deletions are reduced by ~2 fold and gene conversions by ~30 fold compared to a *rad51Δ* single mutant. Altogether these data indicate that Elg1 promotes both Rad51-dependent and Rad51-independent recombination at the RFB. As we know that Rad51-independent recombination depends on Rad52 (*Morrow et al., 2017*; *Nguyen et al., 2015*), we can deduce that Elg1 promotes Rad52-mediated IFSA. We assume that deletions that form in a *rad51Δ elg1Δ* double mutant depend upon residual levels of Rad52 that are able to bind to the collapsed replication fork in the absence of Elg1.

## Conclusion

We have provided evidence that PCNA unloading by Elg1 promotes recombination at a RFB by limiting Fbh1 and Srs2 activity (*Figure 7*). However, several important questions remain unanswered.

Firstly, how does PCNA retention at the RFB lead to an increase in Srs2 and Fbh1 activity? We suspect that PCNA, either directly or indirectly, is needed for the recruitment and/or retention of Fbh1 and Srs2 at the RFB, as has been observed in budding yeast and human (*Bacquin et al., 2013*; *Motegi et al., 2006*; *Papouli et al., 2005*; *Pfander et al., 2005*). Secondly, how is Rad51-independent recombination promoted by Elg1? One possibility is that Srs2 displaces Rad52 from DNA (*De Tullio et al., 2017*) and, thereby, Elg1's role in limiting Srs2 activity at the RFB would promote both Rad51-dependent and Rad51-independent recombination. Thirdly, does PCNA retention at the RFB suppress recombination only by promoting Fbh1 and Srs2 activity? Our data do not rule out the possibility that PCNA influences other processes that impact on recombination. For example, PCNA might directly or indirectly affect fork reversal. Finally, is PCNA unloading by Elg1 actually needed for efficient RDR, or does it simply promote non-allelic/inter-repeat recombination? From our 2DGE analysis there is no apparent deficit in RDR but, as discussed, this method does not necessarily provide a definitive measure of replication restart efficiency. We are currently developing new assays that will enable us to gain a better measure of RDR at *RTS1*.

# Materials and methods

**Key resources table**

| Reagent type (species) or resource | Designation | Source or reference | Identifiers | Additional information |
|---|---|---|---|---|
| Strain, strain background (*S. pombe*) | various strains | *Ahn et al., 2005* | | standard laboratory strain (972) derivatives; see *Supplementary file 2* |
| Strain, strain background (*S. pombe*) | various strains | *Nguyen et al., 2015* | | standard laboratory strain (972) derivatives; see *Supplementary file 2* |
| Strain, strain background (*S. pombe*) | various strains | this paper | | standard laboratory strain (972) derivatives; see *Supplementary file 2* |
| Strain, strain background (*S. pombe*) | various strains | *Lorenz et al., 2009* | | standard laboratory strain (972) derivatives; see *Supplementary file 2* |
| Strain, strain background (*S. pombe*) | various strains | *Morrow et al., 2017* | | standard laboratory strain (972) derivatives; see *Supplementary file 2* |
| Recombinant DNA reagent | pMW777 | this paper | | plasmid; see Materials and methods |
| Recombinant DNA reagent | pST2 | this paper | | plasmid; see Materials and methods |
| Sequence-based reagent | oMW1884 | this paper | | oligonucleotide; see Materials and methods |
| Sequence-based reagent | oMW1885 | this paper | | oligonucleotide; see Materials and methods |

## *S. pombe* strains

*S. pombe* strains are listed in *Supplementary file 2*. The *elg1Δ::natMX6* and *elg1Δ::kanMX6* alleles were obtained from strains supplied by Stuart MacNeill (*Kim et al., 2005*). The *pcn1+::natMX4* strain was made by targeted replacement of the *pcn1+* gene using a DNA restriction fragment from plasmid pMW777, which consists of the *pcn1+* open reading frame next to a *ADH1* terminator and *natMX4* cassette. The *pcn1^{D150E}::natMX4* strain was constructed in the same way as the *pcn1+::natMX4* strain except the DNA fragment was derived from plasmid pST2, which is a derivative of pMW777 in which the *pcn1* gene has been mutated using a Q5 Site-Directed Mutagenesis Kit (New England Biolabs) and oligonucleotides oMW1884 (5'-CATTACTCGAgagTTATTAACTTTGAG-3') and oMW1885 (5'-CGTTGAAATTCGGCAGCA-3').

## Media and genetic methods

Protocols for the growth and genetic manipulation of *S. pombe*, spot assays and assays for recombination have been described (*Jalan et al., 2019*; *Morrow et al., 2017*; *Nguyen et al., 2015*; *Osman et al., 2005*). Recombination experiments were repeated at least three times with between 3 and 10 colonies being assayed in each experiment as described in *Jalan et al. (2019)*. Strains being directly compared were analysed at the same time in parallel experiments. Spot assay plates were incubated at 30°C for 4 days before being photographed. Statistical analysis was performed in GraphPad Prism Version 8.0.2. Due to some of the recombination data not conforming to a normal distribution, comparisons were made using the Mann-Whitney U test. Sample sizes and *p* values are given in *Supplementary file 1*.

## Two dimensional gel electrophoresis

Genomic DNA was prepared from asynchronously growing yeast cultures by enzymatic lysis of cells embedded in agarose and run on 2D gels as described (*Nguyen et al., 2015*). The probe has also been described (*Ahn et al., 2005*).

## Microscopy, image processing and analysis

The method of live cell imaging and analysis has been described previously (*Nguyen et al., 2015*). Mean values in *Figure 5C and D* were compared by the unpaired t-test.

## Acknowledgements

We thank Stuart MacNeill (University of St. Andrews) for the gift of *elg1Δ* strains.

## Additional information

### Funding

| Funder | Grant reference number | Author |
|---|---|---|
| Wellcome | 090767/Z/09/Z | Matthew C Whitby |
| Medical Research Council | MR/P028292/1 | Matthew C Whitby |
| Biotechnology and Biological Sciences Research Council | BB/P019706/1 | Matthew C Whitby |

The funders had no role in study design, data collection and interpretation, or the decision to submit the work for publication.

### Author contributions

Sanjeeta Tamang, Data curation, Formal analysis, Validation, Investigation, Visualization, Methodology, Writing—review and editing; Anastasiya Kishkevich, Data curation, Formal analysis, Validation, Investigation, Writing—review and editing; Carl A Morrow, Formal analysis, Supervision, Validation, Investigation, Methodology, Writing—review and editing; Fekret Osman, Formal analysis, Supervision, Validation, Investigation, Visualization, Methodology, Writing—review and editing; Manisha Jalan, Formal analysis, Validation, Investigation, Methodology, Writing—review and editing; Matthew C Whitby, Conceptualization, Resources, Formal analysis, Supervision, Funding acquisition, Visualization, Methodology, Writing—original draft, Project administration, Writing—review and editing

### Author ORCIDs

Manisha Jalan (iD) http://orcid.org/0000-0002-4467-4934
Matthew C Whitby (iD) https://orcid.org/0000-0003-0951-3374

### Decision letter and Author response

Decision letter https://doi.org/10.7554/eLife.47277.016
Author response https://doi.org/10.7554/eLife.47277.017

## Additional files

### Supplementary files

• Supplementary file 1. Direct repeat recombinant frequencies.
DOI: https://doi.org/10.7554/eLife.47277.011

• Supplementary file 2. *Schizosaccharomyces pombe* strains.
DOI: https://doi.org/10.7554/eLife.47277.012

• Transparent reporting form
DOI: https://doi.org/10.7554/eLife.47277.013

### Data availability

All data generated or analysed during this study are included in the manuscript and supporting files.

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
