## [Decision Letter]

Thank you for submitting your article "The PCNA unloader Elg1 promotes recombination at collapsed replication forks in fission yeast" for consideration by *eLife*. Your article has been reviewed by three peer reviewers, and the evaluation has been overseen by a Reviewing Editor and Detlef Weigel as the Senior Editor. The following individual involved in review of your submission has agreed to reveal their identity: Anna Malkova (Reviewer #1).

The reviewers have discussed the reviews with one another and the Reviewing Editor has drafted this decision to help you prepare a revised submission.

Summary:

This research advance builds upon the authors' previous contributions (Nguen et al., 2015 and Jalan et al., 2019) that 1) established a clever assay to monitor the replication fork restart in *S. pombe*, 2) showed that Rad51 and Rad52 are recruited promptly to the stalled forks, and 3) that the fork restart occurs through the mechanism that is prone to template switching.

Here, the authors asked what happens to the replisome components (e.g. PCNA) when a stalled replication fork transitions into collapsed fork that is re-started by recombination. The authors convincingly demonstrate that the key step in the transition towards recombination at the replication fork is the PCNA unloading from DNA by Elg1. The absence of Elg1 leads to the suppression of recombination-mediated replication promoted by the replication barrier. The authors also demonstrate that in the absence of Elg1, reduced recombination is partially suppressed by deleting *fbh1* or *srs2*, encoding anti-recombination DNA helicases. Based on these data the authors propose that unloading of PCNA by Elg1 limits Fbh1 and Srs2 activity, which in turn promotes recombination responsible for the restart of collapsed replication forks. The reviewers concurred that the data presented in this research advance are scientifically sound and the experiments are well designed, expertly executed and clearly laid out and that no additional experiments are needed to fully support the authors' conclusions. The reviewers also feel that the reported findings provide several novel insights into the molecular mechanisms responsible for the replication restart and therefore are interesting and important. There are a few minor points, however, we would like the authors to address.

Minor points:

1) "However, it should be noted that if fork convergence is faster, and restarted replication is slower, in an *elg1* mutant, then there would be no apparent change in the amount of large Y-shaped and double Y-shaped DNA molecules. Future studies will be needed to clarify exactly what Elg1's role is in promoting TS". This argument is unclear. It will be great if the authors can further clarify this idea.

2) "…in *rad51elg1* double mutant, deletions are reduced by 2-fold". Since the level of deletions remains really high in the double mutant, it will be important to explain how the remaining deletions are formed. Are they formed by the leftover Rad52 that is capable of binding even in the absence of Elg1?

3) Figure 7: The idea is clear, but it appears that it is not well presented by the figure. Basically, it appears that recombination proteins are bound in both left and right scenario, and the difference is represented only by the width of the lines and a slight difference in transparency. A better visual representation is needed.

4) Figure 7: While these are less likely, other possible scenarios may be consistent with the observed dependence on the Elg1-mediated PCNA unloading. For example, the central step in the model involved fork reversal into the chicken foot structure on which the recombination proteins are loaded. Is it possible for PCNA to negatively influence fork reversal itself?

5) The authors report that spontaneous levels of recombination are increased in the *elg1* strain (Figure 1*RTS1-IO*), which is in line with what reported already by Gazy et al., 2013 – G3. It will be informative if the authors make a specific point about the different requirement of Elg1 in their experiments.

6) Please add significance estimation in Figure 3A. In the text, the authors correctly avoid any statement on statistical significance. However, in several places they imply that the effects are relevant. Readers may benefit from statistical analysis.

7) Figure 4. Srs2Δ strain is hyper-rec, as expected, and the *srs2/elg1*Δ strain shows reduced recombination. Authors conclude that deleting *srs2* improves the *elg1* phenotype. One can also look at this as deletion of Elg1 rescues the *srs2*Δ defect. The same is true for Fbh1. Please discuss.

8) Figure 4. Srs2/*elg1*Δ should be sick, as Srs2 and Elg1 are synthetic lethal at least in fission yeast (Gazy et al., 2013 – G3). Could poor growth or cell death of the *srs2/elg1*Δ also account for the rescue? Please discuss.

---

## [Author Response]

Minor points:1) "However, it should be noted that if fork convergence is faster, and restarted replication is slower, in an elg1 mutant, then there would be no apparent change in the amount of large Y-shaped and double Y-shaped DNA molecules. Future studies will be needed to clarify exactly what Elg1's role is in promoting TS". This argument is unclear. It will be great if the authors can further clarify this idea.

We have added further explanation:

“By a process of elimination, these data suggest that Elg1 is required for the TS process itself, although this conclusion assumes that 2DGE analysis provides an accurate measure of RDR efficiency, which may not always be true (see below).

[…] Taking this into account, we cannot exclude the possibility that Elg1 promotes RDR and, in so-doing, drives TS at the 12.4 kb reporter.”

2) "…in rad51elg1 double mutant, deletions are reduced by 2-fold". Since the level of deletions remains really high in the double mutant, it will be important to explain how the remaining deletions are formed. Are they formed by the leftover Rad52 that is capable of binding even in the absence of Elg1?

We have added the following text:

“We assume that deletions that form in a *rad51*∆ *elg1*∆ double mutant depend upon residual levels of Rad52 that are able to bind to the collapsed replication fork in the absence of Elg1.”

3) Figure 7: The idea is clear, but it appears that it is not well presented by the figure. Basically, it appears that recombination proteins are bound in both left and right scenario, and the difference is represented only by the width of the lines and a slight difference in transparency. A better visual representation is needed.

We have revised Figure 7.

4) Figure 7: While these are less likely, other possible scenarios may be consistent with the observed dependence on the Elg1-mediated PCNA unloading. For example, the central step in the model involved fork reversal into the chicken foot structure on which the recombination proteins are loaded. Is it possible for PCNA to negatively influence fork reversal itself?

We have added the following text:

“Thirdly, does PCNA retention at the RFB suppress recombination only by promoting Fbh1 and Srs2 activity? Our data do not rule out the possibility that PCNA influences other processes that impact on recombination. For example, PCNA might directly or indirectly affect fork reversal.”

5) The authors report that spontaneous levels of recombination are increased in the elg1 strain (Figure 1 RTS1-IO), which is in line with what reported already by Gazy et al., – 2013 -G3). It will be informative if the authors make a specific point about the different requirement of Elg1 in their experiments.

We have added the following text:

“In line with the observation that an *elg1*∆ mutant exhibits increased spontaneous direct repeat recombination in budding yeast (Gazy et al., 2013), a fission yeast *elg1*∆ mutant containing *RTS1-IO* exhibits slightly higher levels of spontaneous recombination than a wild-type strain with *RTS1-IO*, suggesting that Elg1 has an anti-recombinogenic function (Figure 1C).”

“In stark contrast to the modest increase in spontaneous recombination, the elevated levels of recombination in wild-type strains containing *RTS1-AO* are reduced dramatically in an *elg1*∆ mutant,….”

“Our data show that Elg1, in addition to having an anti-recombinogenic function, also has a pro-recombinogenic function that is specifically required to promote RFB-induced recombination, which likely explains the reduction in MMS/phleomycin-induced recombination observed previously in budding yeast.”

6) Please add significance estimation in Figure 3A. In the text, the authors correctly avoid any statement on statistical significance. However, in several places they imply that the effects are relevant. Readers may benefit from statistical analysis.

Done.

7) Figure 4. Srs2Δ strain is hyper-rec, as expected, and the srs2/elg1Δ strain shows reduced recombination. Authors conclude that deleting srs2 improves the elg1 phenotype. One can also look at this as deletion of Elg1 rescues the srs2Δ defect. The same is true for Fbh1. Please discuss.

We have added further explanation:

“To investigate this possibility, we compared *RTS1*-induced recombination in a *srs2*∆ single mutant and *srs2*∆ *elg1*∆ double mutant, reasoning that if Elg1’s role in promoting recombination is to limit Srs2 activity, then removal of Srs2 should obviate its need (i.e. a *srs2*∆ *elg1*∆ double mutant should exhibit the same frequency of recombination as a *srs2*∆ single mutant). However, if Elg1’s pro-recombinogenic role is independent of Srs2, then the fold reduction in recombination frequency between *srs2*∆ single mutant and *srs2*∆ *elg1*∆ double mutant should be the same as that between wild-type and *elg1*∆ single mutant.”

“However, the fact that much of the hyper-recombination in a *srs2*∆ mutant is still suppressed by deletion of *elg1* indicates that Elg1’s pro-recombinogenic function is mostly independent of limiting Srs2 activity.”

8) Figure 4. Srs2/elg1Δ should be sick, as Srs2 and Elg1 are synthetic lethal at least in fission yeast (Gazy et al., 2013 – G3). Could poor growth or cell death of the srs2/elg1Δ also account for the rescue? Please discuss.

We have added new data (Figure 4—figure supplement 1 plus associated figure legend) and the following text:

“In budding yeast, a *srs2*∆ *elg1*∆ double mutant exhibits impaired growth and a synergistic increase in sensitivity to DNA damaging agents, including MMS and the Topoisomerase 1 inhibitor camptothecin (CPT), which causes replication fork breakage (Gazy et al., 2013; Parnas et al., 2010). […] Therefore, the differences in recombination between single and double mutant strains is unlikely to stem from a difference in their growth and viability.”

“First, we assessed the viability and genotoxin sensitivity of an *fbh1*∆ *elg1*∆ double mutant (Figure 4—figure supplement 1B). An *fbh1*∆ single mutant exhibits impaired growth relative to wild-type (Osman et al., 2005), which is not exacerbated by deletion of *elg1*. However, an *fbh1*∆ *elg1*∆ double mutant does exhibit a synergistic increase in sensitivity to both MMS and CPT.”